# Active Sampling Techniques for Two-Spot Cotton Leafhopper (*Amrasca biguttula* Ishida) (Hemiptera: Cicadellidae) in Fields with High and Low Populations

**DOI:** 10.3390/insects16121226

**Published:** 2025-12-03

**Authors:** Daphne Zapsas, Susan E. Halbert, Mary Yong Cong, Felipe Soto Adames, Sajan KC, Dakshina Seal, Amy L. Roda

**Affiliations:** 1Science and Technology, Plant Protection and Quarantine, Animal and Plant Health Inspection Service United States Department of Agriculture, Miami, FL 33158, USA; daphne.zapsas@fdacs.gov; 2Division of Plant Industry, Florida Department of Agriculture and Consumer Services, Gainesville, FL 32608, USA; susan.halbert@fdacs.gov (S.E.H.); felipe.soto-adames@fdacs.gov (F.S.A.); 3Division of Plant Industry, Florida Department of Agriculture and Consumer Services, Miami, FL 32608, USA; mary.yongcong@fdacs.gov; 4Department of Entomology and Nematology, University of Florida, Gainesville, FL 32611, USA; sajankc@ufl.edu; 5Tropical Research and Education Center, Institute of Food and Agricultural Sciences, University of Florida, Homestead, FL 33031, USA; dseal3@ufl.edu

**Keywords:** adventive species, cotton jassid, direct sampling, survey methods

## Abstract

Multiple survey techniques are available to detect invasive insect pests. Our study examined eight active sampling techniques to identify the most practical approach for field use to detect a new invasive species, the two-spot cotton leafhopper (*Amrasca biguttula* (Ishida) (Hemiptera: Cicadellidae)). We found that using a tray with a thin layer of 70% isopropyl alcohol consistently captured adult male leafhoppers, which are needed for identification, across both low and high population densities. This method proved reliable regardless of the sampler’s experience, and a large number of plants could be surveyed quickly. This method may provide growers and scouts with a tool to determine if leafhoppers are present in their fields.

## 1. Introduction

Finding a new adventive insect pest as soon as possible after incursion is critical for eradication or timely mitigation efforts to slow the spread of the pest [1]. Many adventive insect surveys rely on passive detection techniques such as sticky traps that intercept flying arthropods [2,3,4]. These are used because of their relatively low service time and the fact that traps remain in operation continuously. However, their use has limitations, as the traps capture the target from unknown distances, and some individuals may not be detected during processing. Small insects, such as leafhoppers (Hemiptera: Cicadellidae), are particularly difficult to detect because of their size [2,4]. They can degrade quickly on a trap or may be missed, particularly when a trap becomes saturated with other non-target insects and debris. In addition, passive traps are not efficient [2,3,4,5], particularly if attractants or pheromones are not available or even known for the target insect. Hence, a target insect’s population will likely be very large before enough individuals are present to be caught randomly on a trap. Alternatively, active collection, as the name implies, involves the direct involvement of the collector, who effectively moves in search of the target. Collectors can use their knowledge of the target to sample preferred hosts directly and select plants that show signs of damage. Active sampling encompasses visual observation, sweep netting, aspirating, beat sheets, pan sampling, vacuuming, and bagging [2,3,4,5,6,7]. The chosen direct sampling method must be sufficiently accurate to determine the presence of the target but also simple and quick enough so that it can be performed frequently and over a large area to allow for timely detection [2,3]. With direct sampling, the more plants that are sampled, the more reliable the sampling data [2,3]. Obviously, there is a trade-off between the number of plants that can be inspected practically and the acceptable level of confidence in the results. In addition, the method used must be repeatable in that it does not require specialized skills or equipment. Finally, the method must be safe for the collector and must limit the exposure of the collector to pesticides if sampling a managed crop. 

We conducted a study to evaluate different direct survey methods to determine the most effective method to detect the two-spot cotton leafhopper (*Amrasca biguttula* (Ishida) Hemiptera: Cicadellidae)) an important cotton (*Gossypium* L.) and okra (*Abelmoschus esculentus* (L.) Moench) pest [8,9,10,11,12]. Native to Asia, the first detection in the Western hemisphere was in Puerto Rico [5], from which it rapidly spread throughout the Caribbean [7]. In the USA, a passive detection strategy was used initially, complemented by the direct methods of visual inspection of host plants and sweep netting [12]. Despite these efforts, the leafhopper was detected first in a highly damaged okra field in the southern tip of the Florida Peninsula in late November 2024 [11]. Subsequent visual surveys revealed that the pest was distributed widely, with a minimal number found in sticky traps. By the end of the week after its initial discovery, the leafhopper had been found in commercial cotton on the border with Alabama and Georgia [11].

Our goal was to find the direct sampling method that was most effective in capturing male leafhoppers, since they are needed for morphological identification [13]. We also determined which method caught females and nymphs effectively, as these individuals can be used for molecular identification if no males are found in the sample, although, molecular testing requires more time and resources. In addition, we evaluated which method was the fastest in capturing the target by evaluating the number of leafhoppers caught per unit time (seconds) per number of plants sampled and explored which method was the most consistent among workers.

## 2. Materials and Methods

### 2.1. Field Sites

We selected a highly infested (ca. 15 leafhoppers/leaf) okra field (60 m × 120 m) in Miami-Dade County, FL, USA (25.415601, −80.515304), to determine which sampling method captured the largest number of total leafhoppers (adults + nymphs), with subtotals for adults and male adults. Okra (*cv.* Clemson spineless) was planted in raised beds covered with black ground cloth. The plants (approximately 1.5 m tall) were at the end of their production cycle (ca. 14 weeks) and showed severe damage (hopper burn) caused by the leafhopper. We determined whether the methods that captured the most leafhoppers were also effective for low populations by sampling a commercial eggplant (*Solanum melongena* L.) field (30 × 400 m) located in Miami-Dade County, FL, USA (25.53547, −80.48638). The eggplants showed no damage, and no two-spot cotton leafhoppers were seen after visually inspecting 10 upper and 10 middle canopy leaves. The eggplant (*cv.* black beauty) was fruiting (approximately 1.5 m tall) and planted in raised beds.

### 2.2. Sampling Methods

Eight sampling methods were tested during this study: (1) aspirating, (2) bagging, (3) beat sheet, (4) sweep netting and tapping the plant material over a tray that was (5) dry or over a tray with (6) alcohol, (7) tap water, or (8) soapy water. Each worker was assigned a row in the field where they performed each of the sampling methods within a 10 m section of the row. Each plot that a worker sampled with a given method was considered a replicate (*n* = 5). After performing the first method, the worker would conduct the next method in the following 10 m of the row. Thus, every sampling method was tested in each row of the field by a different worker and not repeated in the same row. This helped mitigate the potential variance in leafhopper abundance in the field. The study was conducted between 8:00 and 15:00 to reflect the hours that a surveyor or grower likely would be working in the field.

For the aspirating, bagging, and beat sheet methods, the first, middle, and last plants were sampled within the 10 m transect. For the aspirating method, all leafhoppers seen on the leaves and stems were collected with a handheld aspirator, and the vial was capped, labeled, and transported to the laboratory for processing. The bagging method involved enclosing the upper 30–40 cm of a plant inside a clear plastic bag (7.57 L), sealing the stem within the bag, and then tapping the plant three times with a PVC pipe (30 cm) before swiftly removing the bag and retaining all arthropods collected. The bag was sealed, labeled, and transported to the laboratory for processing. For the beat sheet method, a 30.5 × 30.5 cm beat sheet was placed beneath the plant canopy; the plant was tapped three times with the PVC pipe to dislodge arthropods onto the sheet. The worker then gathered leafhopper adults and nymphs with an aspirator.

For the sweep netting and the methods using a tray, all 14 plants in the 10 m transect were sampled. With a fine-mesh sweep net, each worker swept the upper 50 cm portion of the plant canopy, after which the contents were aspirated into a vial. There were four variants of the tray collection method: (1) a dry tray that had no collection liquid, (2) a tray with 70% isopropanol alcohol, (3) a tray with tap water, and (4) a tray with disinfectant soap (GX 1027 antimicrobial soap, diluted 1:54 concentrate/water, Flo-Tec, Clearwater, FL. USA) as a surfactant to reduce the surface tension of the water. For each tray method, a worker placed the plastic tray (25 × 20 cm) beneath the sampled portion of the plant (i.e., top, middle, and bottom) and gently tapped the plant with the PVC pipe so that insects would fall directly onto the trays (Figure 1).

Leafhoppers were aspirated directly off the dry tray. For the methods using a liquid, the bottom of the tray was covered with approximately 50 mL prior to sampling. After sampling, the liquid was poured into a collection vial, and any remaining specimens were collected with a fine paintbrush and placed into the vial with the liquid. Each worker recorded the amount of time that they took to perform each method, which ended when the leafhoppers were contained in a vial or a bag.

### 2.3. Sample Processing and Analysis

In the laboratory, specimens were removed from their vials, and two-spot cotton leafhopper males and females, and leafhopper nymphs were separated and counted using a dissecting microscope (10×). Adult two-spot cotton leafhoppers were identified by their small bright-green bodies, characteristic dark spots on each wing, and two dark spots on their heads when present [8,14]. As no morphological characteristics are available to identify leafhopper nymphs, all potential species were pooled. No adults of any other species were found in the samples. Voucher specimens collected from the study sites were identified as two-spot cotton leafhoppers. Data for the numbers of males, females, and nymphs was analyzed using tests for normality (Shapiro–Wilk test) and homoscedasticity (Bartlett test) with JMP Pro, version18.2 [Cary, NC, USA]. The data were subsequently transformed using log (x + 1) to meet the assumptions of the analysis of variance (ANOVA). The transformed data were subjected to a one-way ANOVA, and pairwise comparisons were made using Student’s *t*-test, with *p* ≤ 0.05 considered statistically significant. We also analyzed the data with a Generalized Linear Mixed Model using the negative binomial distribution (Appendix B) as the design could be considered hierarchical. In both analyses, sample method, stage, and worker were considered fixed effects. Finding similar results (Appendix A), the traditional depiction of data (means and standard error) was selected for simplicity of interpretation. The number of nymphs was compared to the number of adults captured for a given sampling technique using a *t*-test, with *p* < 0.01 considered highly significant and *p* < 0.05 considered significant. The sampling time data did not meet the assumptions of the ANOVA after transformation and were compared using the non-parametric Kruskal–Wallis test at *p* < 0.05.

## 3. Results

In the highly infested okra field, we found that workers collected the most leafhoppers using the tray + alcohol, aspiration, and bagging methods (F_7,38_ = 11.5, *p* < 0.0001, Figure 2). The beat sheet and dry tray method yielded fewer leafhoppers than the tray + alcohol method, but the numbers of two-spot cotton leafhoppers collected with the tray + alcohol method were similar to aspirating and bagging. Interestingly, the sweep netting, tray + soap, and tray + water methods commonly used to collect hemipterans [4] yielded the fewest leafhoppers.

Because adults have more characteristics that help a surveyor recognize the target than nymphs, we looked at which method captures the most adult leafhoppers. There were more adults collected using the tray + alcohol, aspirating, and sweep net methods compared to the numbers of nymphs (Figure 3). The reverse was found for the beat sheet, tray + soap, and tray + water methods, where significantly more nymphs were captured compared to adults (Figure 3). There were no differences in the numbers of adults or nymphs caught using the bagging and dry tray methods.

As the morphological identification of leafhoppers often is based on male genitalia, we looked at which method captured the most male two-spot cotton leafhoppers. We found that the tray + alcohol method captured the most male leafhoppers compared to the other methods (F_7,31_ = 7.63, *p* < 0.0001, Figure 4).

The survey method selected must determine the presence of the target and cover a large area. Three plants were sampled using the aspiration, bagging, and beat sheet methods, while fourteen plants were sampled through sweep netting or dislodging the leafhoppers over a tray. Therefore, we compared the amount of time that each worker took to collect the target from an okra plant. Among the sampling methods tested, the tray + alcohol method was the most time-efficient, while the beat sheet method was the most time-consuming (Figure 5).

In the commercial eggplant field with a low population of leafhoppers, we excluded the aspiration method because of the uncertainty of chemical application within the field site. We also excluded the tray + soap and the tray + water methods because of the low numbers of leafhoppers captured in the okra field using those methods. The bagging, dry tray, and tray + alcohol methods were the only techniques that obtained leafhoppers (Table 1). Male two-spot cotton leafhoppers were collected only using the tray + alcohol method.

## 4. Discussion

Our objective was to evaluate different field sampling techniques to aid in the early detection of the two-spot cotton leafhopper, a new adventive pest in the continental USA [7,8]. Male two-spot cotton leafhoppers are needed for confirmatory identification when molecular identification tools are not available [13,14,15]. Moreover, molecular identification is expensive and takes at least two days, including sequencing, as opposed to about half an hour for genitalic preparation and a dollar for supplies. Male genitalic preparation results in an archivable specimen, whereas molecular identification sometimes does not. The tray + alcohol protocol proved to be a preferred method in that this method captured the most male leafhoppers and was also the only one that obtained a male at the low-density eggplant field site. With the use of the tray + alcohol method, workers were able to sample a large area quickly and safely. This method also was performed uniformly by all personnel regardless of their previous experience with the different direct sampling methods. Interestingly, sweep netting, which is generally considered a good direct sampling method for capturing active plant-dwelling hemipterans [4], was the least effective method used to capture two-spot cotton leafhoppers. We found that the sweep nets easily became tangled in the okra and eggplants and skill was required to keep the net open, particularly in windy conditions.

Using alcohol offered several advantages for collecting these highly mobile leafhoppers. The alcohol likely immobilized or killed the leafhoppers immediately. Interestingly, two-spot cotton leafhoppers were able to avoid capture in water or when a surfactant (disinfectant soap) was added to the water. In fact, using a dry pan proved more effective than the methods with water as hitting the hard surface likely immobilized them so that they could be aspirated. In addition, the use of alcohol facilitated the transfer from field collection to the laboratory for processing. The sample was simply poured into a container, while insects collected on the dry tray, beat sheet, and sweep net had to be aspirated from the collection surface. Finally, the alcohol helps to stop the specimens from degrading, which happens quickly with small insects. This is crucial for preserving specimens for morphological assessment and obtaining quality DNA for genetic analysis [15].

Aspirating is a more targeted approach to detecting two-spot cotton leafhoppers, as a surveyor can choose to search plants showing signs of damage. The aspiration method is low-cost, does not require a large amount of equipment, and is time-effective as the insects are collected directly into a container for transport. We found the aspiration method yielded a significantly larger number of adults versus nymphs and a high average number of males per worker. However, aspirating poses safety concerns if sampling occurs in fields where the use of pesticides is unknown. We also found that when the number of plants surveyed was considered, the aspirating method was not as time-efficient when compared to the tray + alcohol method.

Bagging the plant was found to be an effective method as workers consistently caught a high number of two-spot cotton leafhoppers, including a large number of males. This method is cost-efficient and safer than aspirating in a commercial field. The materials needed are minimal (i.e., a resealable plastic bag that can fit a plant or parts of the plant being sampled). However, bagging might not work in large fields, and this method could stress or damage the plant. Additionally, the number of two-spot cotton leafhoppers caught with the bagging method could be affected by the daily movements of the leafhopper, as has been found with other Cicadellidae species [16,17,18].

## 5. Conclusions

The primary purpose of this study was to find an effective method to detect the two-spot cotton leafhopper in new areas of invasion. This pest has spread rapidly through the USA’s important cotton and vegetable growing regions, and, to date, initial detections have been made when populations are extremely high, often causing severe hopper burn damage [8]. We found that tray + alcohol was the best method for detecting the leafhopper when populations were low and this method was effective for sampling a large number of plants quickly. This method may provide growers and scouts with a tool to determine whether leafhoppers are present in their fields. Additional research is needed to develop and validate sampling protocols that best estimate the pest population size and ultimately provide growers with the information needed to make management decisions.

## Figures and Tables

**Figure 1 insects-16-01226-f001:**
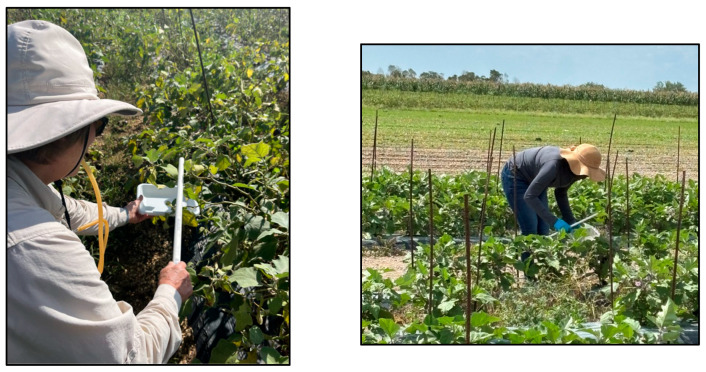
Tray collection method where workers gently tap the host plant over a dry tray and aspirate leafhoppers off the surface or tap the plant over a tray containing a solution.

**Figure 2 insects-16-01226-f002:**
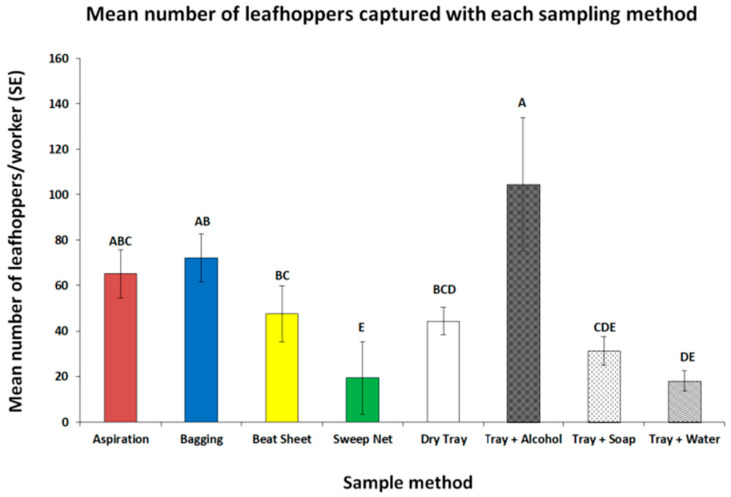
The mean number of leafhoppers (sum of adults and nymphs) per person for each sampling method from a highly infested okra field. Mean values followed by the same letters are not significantly different according to Student’s *t*-test (*p* < 0.05) after a log (x + 1) transformation to meet the assumptions of the ANOVA. Values plotted are back-transformed to the original scale. Bars represent the standard error of the mean.

**Figure 3 insects-16-01226-f003:**
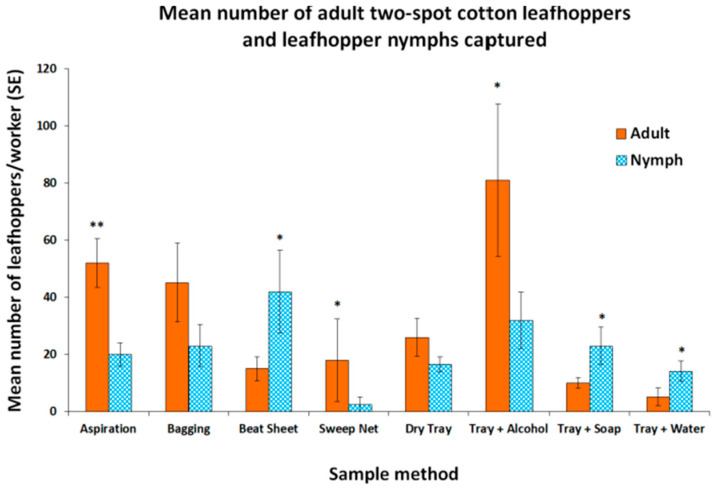
The mean number of two-spot cotton leafhopper (Hemiptera: Cicadellidae *Amrasca biguttula* (Ishida)) adults and leafhopper nymphs collected per worker for each sampling method from a highly infested okra field. An asterisk over the bars indicates a significant difference between the mean number of adults and nymphs caught according to the *t*-test (** *p* < 0.01, * *p* < 0.05) after a log (x + 1) transformation to meet the assumptions of the ANOVA. Values plotted are back-transformed to the original scale. Bars represent the standard error of the mean.

**Figure 4 insects-16-01226-f004:**
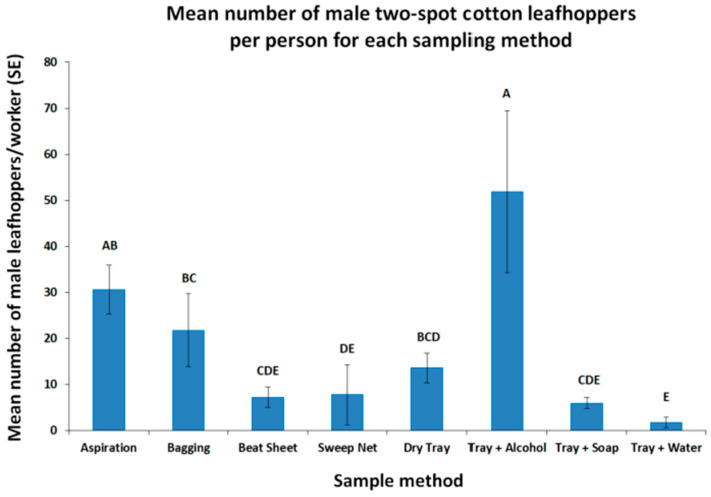
The average number of male two-spot cotton leafhoppers collected per worker for each sampling method from a highly infested okra field. Mean values followed by the same letters are not significantly different according to Student’s *t*-test (*p* < 0.05) after a log (x + 1) transformation to meet the assumptions of the ANOVA. Values plotted are back-transformed to the original scale. Bars represent the standard error of the mean.

**Figure 5 insects-16-01226-f005:**
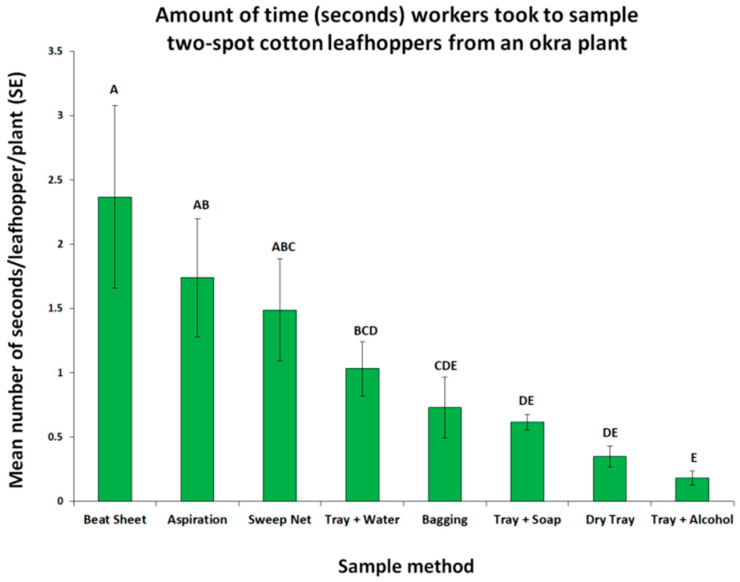
The efficiency of each sample method based on the average amount of time (seconds) that workers took to collect two-spot cotton leafhoppers (*Amrasca biguttula* (Ishida) Hemiptera: Cicadellidae) adults and nymphs per okra plant sampled. Mean values followed by the same letters are not significantly different according to the Kruskal–Wallis test (*p* < 0.05). Bars represent the standard error of the mean.

**Table 1 insects-16-01226-t001:** The mean number of two-spot cotton leafhoppers (±standard error of the mean) collected by each worker from a commercial eggplant field with a low population of leafhoppers.

Sample Method	Adults + Nymphs Leafhoppers/Worker	Adult Leafhoppers/Worker	Male Leafhoppers/Worker
Bagging	0.4 (0.24)	0.2 (0.2)	0
Beat sheet	0	0	0
Netting	0	0	0
Dry tray	0	0	0
Tray + Alcohol	1 (0.44)	1 (0.55)	0.8 (0.37)

## Data Availability

All data are included in the main text.

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
