# Peer review of "Active Sampling Techniques for Two-Spot Cotton Leafhopper (Amrasca biguttula Ishida) (Hemiptera: Cicadellidae) in Fields with High and Low Populations"

_insects, 2025, doi:10.3390/insects16121226_

Round 1

Reviewer 1 Report

Comments and Suggestions for Authors

The manuscript "Active sampling techniques for the two-spotted cotton leafhopper (Amrasca biguttula Ishida) (Hemiptera: Cicadellidae) in fields with high and low populations" is based on valid data, but they have been sub-optimally reported and, in my opinion, require thorough revision.

Below are my comments

  • Lines 43–51. This part of the introduction must be supported by references. In any case, the efficiency of sticky traps varies depending on the species sampled and it is not possible to generalize. The criticism of the use of traps should be made in reference to Amrasca biguttola as reported in lines 70-71, citing the study that showed this.
  • Lines 63–78. In this part of the introduction section, the aim of the study is not clearly stated. As clearly stated in lines 39-40, 201, 242-243, the aim is not “to determine the most effective method to detect a small insect in a crop”. The fact that the insect is small is not important. Also, I'm not entirely sure that the term "effective" is the most correct one, if this does not refer to the same unit of sampling time. Therefore, I would replace the sentence with " The aim is to determine the most efficient sampling method for detecting biguttola when it is highly dispersed” (i.e., the sampling method that obtains the maximum information with the minimum sampling time; i.e., that allows, for the same amount of time, the greatest probability of finding the insect when it is highly dispersed).
  • Lines 81 and 88. I do not consider it optimal to have used two different host plants, as the plants could influence the sampling methods compared differently.
  • Line 104–113. Regarding the "suction method", there are two issues: (i) were the collected individuals seen randomly on the upper leaf surface/shoot axils or by turning the leaves? (ii) Leafhoppers have a daily periodicity, so they normally remain still and hidden during the hours of sunlight and move at dawn and dusk, so the time at which this sampling was carried out is crucial.

See, for example, three studies on tribe Empoascini:

  • Smith S. M., Ellis C. R., 1983. Economic importance of insects on regrowths of established alfalfa fields in Ontario. Canadian Entomologist, 115: 859-868.
  • Taylor R. A. J., Reling D., 1986. Preferred wind direction of long-distance leafhopper (Empoasca fabae) migrants and its relevance to the return migration of small insects. Journal of Animal Ecology, 55: 1103-1114.
  • Pavan F., Cargnus E., Zandigiacomo P., 2023. Use of yellow sticky traps to study daily flight activity and behaviour of sap-sucking insects inhabiting European vineyards. Bulletin of Insectology, 76: 101-115.

For the "bagging method," the time of day is also crucial, as during the hours when they are most active, they are more likely to escape while bagging the shoots. For the "sweep netting method," the time of day is also crucial, as more insects are caught when they move outside the canopy than when they are sheltered.

  • Line 118. Are you sure that isopropanol alcohol doesn't cause health problems?
  • Lines 134–136. There is much talk about the need to collect males for species identification, but why are all specimens identified based on external morphological characters? If there are no species that can be confused, it makes no sense to talk about the need to capture males for species identification and say that PCR is necessary for nymphs and females. At line 155, it says: “The nymphal stage of leafhoppers has few morphological characters to distinguish species reliably”. Consequently, in Materials and Methods, it would be necessary to say how the nymphs were identified and, if so, with which species they can be confused.
  • Line 136. It might be useful to point out that the replicates are represented by the 6 workers.
  • Figure 2. The data reported in Figure 2 does not make sense based on the aim of this study, since the best sampling method must be the one that allows the greatest catches per unit of time.
  • Figures 3 and 4. In Figures 3 and 4, instead of reporting the total catches, given that the sampling times are not the same, it would be more interesting to report, respectively, adults/nymphs ratios and males/females ratios.
  • Figure 5. I'm not clear on the reference to the plant rather than just the unit of time. To compare the efficiency of sampling methods—that is, which method captures more individuals per unit of time—the number of plants sampled is irrelevant. From a practical point of view, what matters is knowing which sampling method, in the presence of dispersed populations, guarantees the greatest probability, for the same sampling time, of capturing more individuals per unit of time. If the number of plants sampled varies, it doesn't matter.
  • Lines 191–194. This information should be reported in the materials and methods.
  • Lines 205-206. In your study, you have identified both males and females, both adults and nymphs, based on external morphology. Therefore, once again, I don't understand why you talk about the need for males.
  • Lines 223–225. I don't understand the need to preserve DNA if identification can be made based on external morphology.
  • Discussion and Conclusion. I believe that the discussion should be based primarily on the most efficient method, that is, the one that allows the greatest number of individuals to be captured per unit of time. It might also be interesting to discuss the ratios of nymphs to adults and males to females.

Author Response

The authors appreciate the time and effort that Reviewer 1 took to make a careful and detailed review. The manuscript has improved by addressing the reviewer's concerns.

The manuscript "Active sampling techniques for the two-spotted cotton leafhopper (Amrasca biguttula Ishida) (Hemiptera: Cicadellidae) in fields with high and low populations" is based on valid data, but they have been sub-optimally reported and, in my opinion, require thorough revision.

Below are my comments

  1. Lines 43–51. This part of the introduction must be supported by references. In any case, the efficiency of sticky traps varies depending on the species sampled and it is not possible to generalize. The criticism of the use of traps should be made in reference to Amrasca biguttulaas reported in lines 70-71, citing the study that showed this.

We agree that the referred portion of the introduction must be supported by references. We have added the reference below that we used to frame the introduction, but inadvertently omitted. These references highlight that we concur with the reviewer that the use of sticky traps varies depending on species. These references also highlight that they are not the preferred method for all species. Additionally, they indicate that the use of sticky traps depends on the purpose.  For example, sticky traps might not be a good tool to detect a leafhopper in a new area of invasion, but they may serve to monitor the species once it is established.  For A. biguttula, sticky traps are not a good option. The insects decay, making male genitalic preparations challenging and molecular analysis impossible. Moreover, our observations indicate that this species is not attracted to yellow, the most common color for the traps. Like sticky cards, some direct sampling methods may not be an effective approach to sampling a target pest. We use the first paragraph of the introduction to broadly frame the approaches taken to survey for an insect to build the background for why we looked at different direct sampling methods for A. biguttula, which follows in the next paragraph.

References added to the introduction:

Disney, R.H., Erzinclioglu, Y.Z. (1982). Collecting methods and the adequacy of attempted fauna surveys, with reference to the Diptera. Field Studies. 5(4):607-21.

Grootaert, P., Pollet, M., Dekoninck, W., van Achterberg, C. (2010) Sampling insects: general techniques, strategies and remarks. In Eymann, J., Degreef, J., Hāuser, J. Monje, C., Samyn, Y. and Vanden Spiegel, D.(eds.) Manual on Field Recording Techniques and Protocols for All Taxa Biodiversity Inventories and Monitoring. Abc Taxa, Belgium. 337-99.

Schwertner, C.F., Carrenho, R., Moreira, F.F.F., Cassis, G. (2021). Hemiptera Sampling Methods. In: Santos, J.C., Fernandes, G.W. (eds) Measuring Arthropod Biodiversity. Springer, Cham. https://doi.org/10.1007/978-3-030-53226-0_12

  1. Lines 63–78. In this part of the introduction section, the aim of the study is not clearly stated. As clearly stated in lines 39-40, 201, 242-243, the aim is not “to determine the most effective method to detect a small insect in a crop”. The fact that the insect is small is not important. Also, I'm not entirely sure that the term "effective" is the most correct one, if this does not refer to the same unit of sampling time. Therefore, I would replace the sentence with " The aim is to determine the most efficient sampling method for detecting biguttolawhen it is highly dispersed” (i.e., the sampling method that obtains the maximum information with the minimum sampling time; i.e., that allows, for the same amount of time, the greatest probability of finding the insect when it is highly dispersed).

We agree with the reviewer that our study was focused on A. biguttula over small insects in crops. We also agree that ‘effective’ may include specific technical aspects of sampling that we did not test in this study. Therefore, we have changed the wording in the introduction and throughout the manuscript to better state our goal of testing different types of direct sampling methods to determine how many A. biguttula each method captured. We have also reviewed and modified the text to remove suggestions that we were evaluating the efficacy as it refers to insect population sampling. This was not our goal. We wanted to test different methods and determine which method found the most A. biguttula, which method was likely to include a male, and which methods were repeatable/consistent among works.  There are established leafhoppers that superficially resemble A. biguttula, so diagnostic male genitalic preparations are needed for species identification, especially for regulatory purposes. When available, molecular tools can be used on females and other life stages. Therefore, we also looked at the number of females and nymphs caught as molecular tools could be used to confirm their identification. Our goal was to give surveyors, scouts, and growers methods to find and determine A. biguttula was present. Our aim in terms of evaluating the time it too perform a method was basic in that we want to provide a preliminary evaluation of the amount of time that the method took. We did not have the intention to provide the most efficient method. Our main goal was to look at the quality of the sample in term of how the sample could be used to make confirmatory identification that A. biguttula was present in the field.

  1. Lines 81 and 88. I do not consider it optimal to have used two different host plants, as the plants could influence the sampling methods compared differently.

We chose to use two different host plants for the reason stated by the reviewer (i.e., we wanted to look at how the sampling method is affected by crop type). We wanted to test whether the protocols we were independent of crop type. As this is a new pest, we do not know which host plants will be the one used by the insect. In addition not all host plants are grown in all regions. Using multiple crops allowed us to see that, despite the differences, the tray + alcohol method proved to be the best to find A. biguttula, showing a wider applicability of the method. Thus, we believe our studies provide useful information and a step in understanding how to detect this pest. We do agree that future studies that include controlled studies evaluating the methods in the same crop with different population levels would be very valuable. These studies could also be paired with understanding how the sampling method relates to the population of the leafhopper.

Recently, Florida Department of Agriculture and Consumer Services Division of Plant Industry (FDACS-DPI) inspectors have been asked to sample ornamental plants for sale for A. biguttula. Unfortunately, little training was provided. Samples sometimes were not usable at all, and sometimes we received a single female or a few nymphs, which are not determinable morphologically for regulatory purposes. Molecular analysis takes several days and costs much more than a male genitalic preparation.  After training in the tray + alcohol method, samples improved greatly, and few specimens had to be forwarded to the Molecular Diagnostic Laboratory.

  1. Line 104–113. Regarding the "suction method", there are two issues: (i) were the collected individuals seen randomly on the upper leaf surface/shoot axils or by turning the leaves? (ii) Leafhoppers have a daily periodicity, so they normally remain still and hidden during the hours of sunlight and move at dawn and dusk, so the time at which this sampling was carried out is crucial.

See, for example, three studies on tribe Empoascini:

Smith S. M., Ellis C. R., 1983. Economic importance of insects on regrowths of established alfalfa fields in Ontario. Canadian Entomologist, 115: 859-868.

Taylor R. A. J., Reling D., 1986. Preferred wind direction of long-distance leafhopper (Empoasca fabae) migrants and its relevance to the return migration of small insects. Journal of Animal Ecology, 55: 1103-1114.

Pavan F., Cargnus E., Zandigiacomo P., 2023. Use of yellow sticky traps to study daily flight activity and behaviour of sap-sucking insects inhabiting European vineyards. Bulletin of Insectology, 76: 101-115.

For the "bagging method," the time of day is also crucial, as during the hours when they are most active, they are more likely to escape while bagging the shoots. For the "sweep netting method," the time of day is also crucial, as more insects are caught when they move outside the canopy than when they are sheltered.

For the aspiration/suction method, each worker was given instructions to collect all leafhopper adults and nymphs they found on the entire plant. The instructions included inspecting the top surface and bottom surface of each leaf. Leafhoppers were likely disturbed and lost in the process of inspection. The number of leafhoppers lost likely varied and was based on each worker’s skill in sampling this type of insect. Factors such as the workers past experience with leafhopper behavior, speed and agility, eyesight, and ability to inhale (deeply and quickly) likely affected if a leafhopper escaped or not.  We specifically tested multiple workers with multiple levels of skill in sampling leafhoppers and physical abilities. We hoped that this would give an indication of the consistency of an A. biguttula would be found using each method.

We agree with the reviewer that the time of day may affect the activity of the leafhoppers and this activity might affect the number that was caught. Further studies are needed to determine if A. biguttula exhibits this behavior and how this behavior relates to population, crop type, season, etc. and how any differences in activity affects the number caught with each sampling method.  In our study we tried to simulate a surveyor looking for this new pest and the situation (i.e. the field setting and the time of day) that would be typical for the worker. Thus we evaluated each method during these likely working hours. Our purpose was not to give the final evaluation of efficiency or accuracy of a particular method and whether the method would be improved with time of day. Rather, we wanted to provide growers and surveyors a robust tool with which they were most likely to find the stage that would confirm the pest was present, preferably when A. biguttula numbers were very low.  We have added details to the methods describing the time when we conducted the studies as well as included details in the discussion on how the bag method catch may be affected by the daily movements of the leafhopper.

  1. Line 118. Are you sure that isopropanol alcohol doesn't cause health problems?

There could be a potential for isopropanol alcohol to cause health problems if ingested. Additionally, isopropanol alcohol could cause moderate irritation of the eyes, nose, and throat if a worker is exposed high concentrations of vapor. However, the handling of isopropanol alcohol during this study occurred in an open, outdoor setting. Additionally the amount of alcohol used to cover the tray was minimal (ca. 50 ml) further limiting risk to the workers.  Proper labels (expected) would reduce the likelihood of ingestion.

  1. Lines 134–136. There is much talk about the need to collect males for species identification, but why are all specimens identified based on external morphological characters? If there are no species that can be confused, it makes no sense to talk about the need to capture males for species identification and say that PCR is necessary for nymphs and females. At line 155, it says: “The nymphal stage of leafhoppers has few morphological characters to distinguish species reliably”. Consequently, in Materials and Methods, it would be necessary to say how the nymphs were identified and, if so, with which species they can be confused.

Although A. biguttula adults have characteristics black spots on the wings and frequently on the head that help identify this species, there are several taxa established in the U.S.A. that exhibit phenotypic similarities, potentially leading to misidentification. These include the following species:

  • Empoasca nymphs closely resemble those of A. biguttula, while adults are comparable in size and shape but lack the characteristic black spots present in A. biguttula. There are over 50 species of Empoasca (https://hoppers.speciesfile.org/otus/42128/overview), but as far as we know, none have spots.
  • Typhlocyba pomariaindividuals are light green and similar in size, yet they do not possess the distinctive black markings. https://hoppers.speciesfile.org/otus/48251/overview
  • Kyboasca adults are similar in size to A. biguttulaand bear a black spot on each forewing; however, they lack spots on the anterior margin of the head. These are the most similar to A. biguttula. About half of the species occur only in Central Asia, but at least two occur in the USA and possess similar wing spots. The white markings on the head, and the male genitalia are totally different, as are host plants. Although unlikely, there is a possibility that one of these hoppers might be collected in a trap or randomly on a plant. This is the reason for the need for male for male genitalic preparations or molecular analysis of a female specimen to confirm a record or for implementing regulatory actions. Nymphs cannot be identified to species in any case. https://hoppers.speciesfile.org/otus/1131369/overview.
  • Alconeura adults display black dots on both forewings but also have various colored markings on the wings and lack anterior head markings. They are not green in life.

For confirmatory and regulatory taxonomic resolution, male specimens are needed due to the presence of morphological traits in the genitalia that are absent in females and nymphs. This is typical for identification of leafhoppers and other Auchenorrhyncha. As molecular tools might not be immediately available to growers and decision makers, finding a male for morphological identification could be imperative.  Moreover, molecular diagnosis takes at least two days, including sequencing, and it is expensive. In fact, morphological identification of A. biguttula prompted the immediate mobilization of the Florida Department of Agricultural and Consumer Services while they waited for molecular confirmation. For these reasons, we focused on finding a male leafhopper.  Amrasca biguttula specimens collected from the highly infested okra field used in our study were confirmed both morphologically and genetically.  We learned to recognize A. biguttula adults from other species based on morphological characteristics of size, color, and spots on the wings and head.  We found no other species in our samples.  We did not genetically confirm the identification of the nymphs that were collected in this study.  Although we found only A. biguttula adults, there exists the possibility that a few nymphs were of other species, Empoasca spp., for example. Our goal with this manuscript was not to determine the population of A. biguttula nymphs in relation to adults. Rather, our goal was to show which method collected the most leafhopper nymphs that could be used for molecular analysis if no other specimens were available. We have added to the results section that specimens were identified morphologically and genetically to be A. biguttula and that all adults collected exhibited the morphological characteristics of A. biguttula. We added to the material and methods section that we were not able to identify the nymphs to species. We included the following references that provide a list of the similar species to A. biguttula and highlight the importance of finding a male. 

PPQ. Cooperative Agricultural Pest Survey (CAPS) Pest Datasheet for Amrasca biguttula (Cicadellidae): Cotton jassid. United States Department of Agriculture, Animal and Plant Health Inspection Service, Plant Protection and Quarantine (PPQ), Raleigh, NC.  2025. Amrasca-Amrasca-biguttula-CAPS-Datasheet-for-PDF.pdfbiguttula-CAPS-Datasheet-for-PDF.pdf (accessed 2 October 2025)

Cabrera-Asencio, I., Dietrich, C. H., & Zahniser, J. N. (2023). A new invasive pest in the Western Hemisphere: Amrasca biguttula (Hemiptera: Cicadellidae). Fla. Entomol., 2023. 106(4), 263-266.

  1. Line 136. It might be useful to point out that the replicates are represented by the 6 workers.

We have indicated in the methods that a worker was considered a replicate.

  1. Figure 2. The data reported in Figure 2 does not make sense based on the aim of this study, since the best sampling method must be the one that allows the greatest catches per unit of time.

Collecting as many leafhoppers as possible for the effort (time) was an important aspect of our study. However, our primary goal was to confirm the presence of A. biguttula (or not). Confirmatory identification requires a male specimen or access to molecular identification tools if only females or nymphs are collected. We feel that showing the total number of leafhoppers that each method collected is important as it indicates which method collected the most A. biguttula.  In subsequent graphs, we further evaluate the quality of each sample by examining the composition in terms of the number of males, adults, and nymphs collected. We found that certain methods yielded large numbers of immature leafhoppers, which cannot be identified based on morphology.  Using the sampling techniques that resulted in a larger number of nymphs and perhaps no adults collected would not support the timely decision-making needed for growers or regulators unless genetic identification is available.

  1. Figures 3 and 4. In Figures 3 and 4, instead of reporting the total catches, given that the sampling times are not the same, it would be more interesting to report, respectively, adults/nymphs ratios and males/females ratios.

We chose to show the average number caught and the variance, as this provides an indication on how consistently a method was performed among workers. We found that there were large differences in the number of leafhoppers caught with some methods, which was likely due to the skill level of the sampler. For example, 5 of the 6 workers caught less than 3 leafhoppers using a sweep net while one worker caught 99 leafhoppers. As most workers caught the lowest number of leafhoppers with this technique, we would not recommend it over the others unless a worker was skilled at using a sweep net. Although providing the result as ratios will help reporting the trap catches given the sampling times are not same, we feel that an important piece of information would be lost that could be used to decide what type of sampling method to instruct surveyors to use. The method that collects the most leafhoppers per unit time, might not necessary be the method that satisfies the other goals of this study (i.e. collection of a specimen that could be used to confirm the presence of the pest and a method that can be performed with similar results based on a worker’s skill level). 

  1. Figure 5. I'm not clear on the reference to the plant rather than just the unit of time. To compare the efficiency of sampling methods—that is, which method captures more individuals per unit of time—the number of plants sampled is irrelevant. From a practical point of view, what matters is knowing which sampling method, in the presence of dispersed populations, guarantees the greatest probability, for the same sampling time, of capturing more individuals per unit of time. If the number of plants sampled varies, it doesn't matter.

We considered sampling as many plants as possible an important factor as we do not yet know how A. biguttula will invade fields and without this knowledge wanted a technique that would cover a larger area.  In our study, three plants were sampled using the aspiration, bagging, and beat sheet techniques while 14 plants were sampled using a sweep netting and the three methods with the tray.  Figure 5 allows the reader to see a relationship between the number of leafhoppers caught and the area in terms of number of plants sampled.

  1. Lines 191–194. This information should be reported in the materials and methods.

We added to the materials and method section that we used a t-test to compare the difference in the number of adults and nymphs capture for each method.

  1. Lines 205-206. In your study, you have identified both males and females, both adults and nymphs, based on external morphology. Therefore, once again, I don't understand why you talk about the need for males.

Male A. biguttula are needed for confirmatory identification when molecular identification tools are not available.   In our study, we used size, color, and spots on the wings and head to identify A. biguttula adults in our collections, once the identification of A. biguttula was confirmed morphologically and genetically.  We did not use morphological characteristics to identify the nymphs. We assumed that the nymphs were A. biguttula because no other leafhopper specie was found in our samples. The aim of our study was to address the issue that collecting nymphs will not provide a grower or decision maker the information they need (i.e. whether A. biguttula is present) unless molecularly identification tools are readily available. If the molecular tools are not readily available, a method that would more likely capture a male would be preferred over a method that collects nymphs.

  1. Lines 223–225. I don't understand the need to preserve DNA if identification can be made based on external morphology.

Confirmatory morphological identification of A. biguttula can only be made using male genitalia. Any females or nymphs collected would need to be preserved for molecular analysis for regulatory purposes or to establish a record. We highlighted this in the introduction where we stated this issue as well as provided the supporting reference. We have modified the manuscript to clarify that although A. biguttula has characteristics that can help separate adults from other species they cannot be used to confirm species identification. We also clarified that we did not use morphological characteristics to separate the nymphs. In addition, we have included the reference that describes the similar looking species of leafhoppers that may be encountered as this shows the difficulty in relying on external morphology for adults.  

  1. Discussion and Conclusion. I believe that the discussion should be based primarily on the most efficient method, that is, the one that allows the greatest number of individuals to be captured per unit of time. It might also be interesting to discuss the ratios of nymphs to adults and males to females.

As discussed above, our aim in this study was broader than finding the most efficient method in terms the number of individuals captured per unit time. Our main goal was to find the method that would help a surveyor or grower confirm that A. biguttula was present. We also wanted to look at what method was consistent in term of leafhoppers captured, as this would provide an indication that specialized skills, knowledge or physical abilities were not needed to obtain the leafhoppers. Capturing the largest number of leafhoppers in the least amount of time was an important consideration added with the others factors we evaluated. We have modified the text to better explain the issues using morphological characteristic for this species and that morphological characteristics are not available to distinguish nymphs.  With the clarification of the issues with morphologically identify this specie and clarification of our goals, by incorporating the review’s suggestions, we believe that reader will find the discussion and conclusion appropriate.

Reviewer 2 Report

Comments and Suggestions for Authors

...

Author Response

I found it simple and well written paper. It was easy reading. The objective is straight forward. I have a few issues to consider, though.

1. For the aspirating method, all leafhoppers seen on the leaves and stems were collected with a handheld aspirator, the vial capped, labeled, and transported to the laboratory for processing. Were they not disturbed?

Whether the leafhoppers were disturbed and lost in the process of inspecting the plant likely varied based on each worker’s skill in sampling this type of insect. Factors such as the workers past experience with leafhopper behavior, eyesight, and speed likely affected if a leafhopper escaped or not.  We specifically tested multiple workers with multiple levels of skill in sampling leafhoppers as we wanted to see how this would affect the likelihood a leafhopper would be captured with a particular method. Our aim was to find a method that was the most consistent/repeatable among the workers. 

2. Transformation implies assumptions did not hold. If that was the case, it should be stated right here in the data analysis section

We edited the text to clearly state that our data did not meet the assumptions of ANOVA and therefore had to be transformed before analysis.

3. Are the data here transformed values or back-transformed? It should be clearly stated.

The data shown in Fig. 2-4 are the back-transformed values. The manuscript has been update to state that this was done.

4. Figure 3 includes Figure 2, so Fig 2 may not be necessary. The same title and everything. I think Figure 2 shows sums of adults and nymphs. Also, transformed or raw?

Figure 2 shows, as the reviewer noted, the sum of adults and nymphs collected. We feel this graph provides the reader an easier way to compare the number of leafhoppers caught with each of the method  as well as the variance (the differences between worker). Figure 2 allows the readers to easily see which methods caught statistically similar numbers of leafhoppers by looking at letters over the bars.  Figure 3, although showing similar data, provides the reader with a visual to easily compare the number of leafhopper nymphs collected to the number of A. biguttula adults.  In addition, data in each graph was analyzed differently. Figure 2 an ANOVA was conducted on the transformed data followed by a Student’s t test comparison of each sample. In Figure 3, the number of nymphs was compared to the number of adult A. biguttula capture for a particular method (i.e. the number of nymphs captured using the aspiration method was compared to the number of A. biguttula captured using the aspiration method. In Figure 3, the number of nymphs caught with one method was not compared to the number of nymphs with another method (i.e. nymphs caught using the aspiration method were not compared to the number of nymphs caught using the bag method). We believe these differences warrant both graphs be retained in the manuscript. The titles have been modified to better indicate their differences.

5. Should not the sampling time be uniform all over? If it means the time taken for the worker to see the first speciment drop on the tray, then I can understand. By the way, how long does it take shaking the plant on to the tray three times with a PVC?

The sampling would not necessarily be uniform between the methods or between the individual workers.  We based sampling time on how long it took a work to perform the method until the sample was contained.  Each method had different steps that would likely cause the sampling time to be longer. For example the aspiration method the work collected the specimens directly into a collection vial, thereby foregoing the step (and time) it took the worker to pour the alcohol into a collection vial.  With the alcohol + tray method, the worker would forgo the time spent to carefully search the plant visually. Shaking the plant over a tray would take a work 2-3 seconds, which was much less than the >15 that took some workers to visually inspect a plant.  Additionally, there was variability between workers.  As mentioned above, factors such as the workers past experience with leafhopper behavior, eyesight, and speed likely affected how long it took them to sample a plant.